

# The potency of minerals to reduce oriental fruit fly infestation in chili fruits

Josua Crystovel Pangihutan, Danar Dono and Yusup Hidayat

Department of Plant Pests and Diseases, Universitas Padjadjaran, Sumedang, West Java, Indonesia

## ABSTRACT

**Introductions:** In many areas, particularly in Asia, the oriental fruit fly *Bactrocera dorsalis* (Hendel) causes considerable fruit damage on various plants. The fruit fly causes significant economic losses every year due to reduced fruit quantity and quality as well as export restrictions. This study aimed to examine the potency of minerals in controlling the oriental fruit fly infestation in chili fruits.

**Methods:** Experiments were conducted under laboratory and semi-field conditions using randomized block design. Ten minerals (*i.e.* kaolin, talc, zinc oxide, bentonite, sulfur, dolomite, calcium oxide, calcium hydroxide, calcium carbonate, and zeolite) and an untreated control were tested under laboratory conditions. Twenty chili fruits at a green stage were soaked in each mineral suspension (2%, w/v), air-dried, and placed in a trial cage (23-L plastic container) containing 20 female oriental fruit flies. In a semi-field bioassay using a screen cage (100 cm × 70 cm × 120 cm), 20 female oriental fruit flies were exposed to a fruit-bearing chili plant sprayed with mineral suspension.

**Results:** Talc and calcium oxide significantly reduced the numbers of visiting fruit flies, oviposition holes, and eggs laid, as well as the percentage of infested chili fruits in a laboratory bioassay. Calcium hydroxide was substantially better than talc in controlling fruit fly infestation in a semi-field bioassay, although it was not significantly different from calcium oxide and calcium carbonate.

**Conclusion:** Overall, calcium oxide is a viable option for the long-term control of the oriental fruit fly on chili fruits. Calcium oxide could be utilized as the push component of a push-pull strategy to manage oriental fruit fly infestation in chili fruits because of its potential to inhibit the number of visiting fruit flies and oviposition.

# INTRODUCTION

The oriental fruit fly, *Bactrocera dorsalis* (Hendel) (Diptera: Tephritidae), is a fruit fly species that causes severe fruit damage in Asia. This fruit fly is a polyphagous pest that feeds on various fruits and vegetables (*Siderhurst & Jang, 2006*). *B. dorsalis* has been aligned with the former species *Bactrocera invadens*, *Bactrocera papayae* Drew & Hancock, and *Bactrocera philippinensis* Drew & Hancock (*Schutze et al., 2015, 2017*).

The oriental fruit fly causes significant economic losses every year due to reduced fruit quantity and quality, as well as export restrictions (*Wei et al., 2017*). One of the fruits

Corresponding author
Yusup Hidayat,
yusup.hidayat@unpad.ac.id

frequently affected by oriental fruit flies is chili (*Capsicum annuum* var. *annuum* L.). Fruit fly trap, fruit bagging, and synthetic insecticide application are the standard methods used by farmers to control fruit flies. However, these control techniques have limits as the fruit fly trap still relies on the male attractant (methyl eugenol). Due to the tiny size and number of chili fruits to be bagged per plant, fruit bagging would be inefficient. Synthetic pesticides can cause insect resistance, non-target organism death, and pesticide residues in agricultural products (*Sivaperumal, Anand & Riddhi, 2015*; *Beers et al., 2016*; *Guedes, 2017*). As a result, better solutions for oriental fruit fly management on chili fruits are required.

Minerals are widely utilized as crucial raw material or additives in cement, paint, ceramics, pharmaceuticals, cosmetics, and agrochemicals manufacturing (*Pruett, 2016*; *Moraes et al., 2017*; *Wang et al., 2018*; *Viseras et al., 2019*; *Manjaiah et al., 2019*). Minerals are one of the raw materials used in agriculture to make fertilizers and pesticides. Minerals are frequently used as carriers in manufacturing fungicides and insecticides in the pesticide industry (*Yusoff, Kamari & Aljafree, 2016*). Minerals have also been documented as fungicides in treating plant diseases (*Jamar et al., 2017*). Except for kaolin, little is known regarding minerals for insect control.

Kaolin has been used to control Mediterranean fruit flies, *Ceratitis capitata* (Wiedemann) (*Mazor & Erez, 2004*; *D'Aquino et al., 2011*; *Lo Verde, Caleca & Lo Verde, 2011*; *Campos Rivela & Martínez-Ferrer, 2013*), olive fruit fly, *Bactrocera oleae* (Rossi) (*Caleca & Rizzo, 2007*; *Pascual et al., 2010*; *Mozhdehi & Kayhanian, 2014*), cherry fruit fly, *Rhagoletis cerasi* L (*Mezőfi et al., 2018*), *Anastrepha fraterculus* (Wied.) (*Ourique et al., 2019*), and spotted wing drosophila fruit fly, *Drosophila suzukii* Mats. (*Knapp, Mazzi & Finger, 2019*). Kaolin can also be used to suppress the attack of other insect pests such as leafminer *Liriomyza huidobrensis* (Blanchard), aphid *Myzus persicae* Sulzer (*Soubeih, Ali & El-Hadidy, 2017*), boll weevil, *Anthonomus grandis* Boheman (*Silva & Ramalho, 2013*), leafhopper *Empoasca vitis* (Göthe) (*Markó et al., 2008*), and cotton bollworm, *Helicoverpa armigera* (Hübner) (*Alavo et al., 2011*). Despite this, little is known about the use of kaolin for the control of the oriental fruit fly infestation.

Efforts to employ minerals in crop protection may have a bright economic future. This effort also aligns with the community growing demand for safer plant protection products. Mineral origin pesticides have benefits over other natural pesticides, such as botanical and microbiological pesticides, in terms of raw material availability and natural persistence. Minerals such as kaolin are made up of particles not damaged by high heat or direct sunshine (*Braham, Pasqualini & Ncira, 2007*). Female oriental fruit flies lay their eggs beneath the skins of fruits, and the larvae hatch and develop inside the fruits. This position makes it challenging to control the larvae, even with synthetic insecticides. As a result, it is vital to avoid or limit the damage by preventing the oriental fruit flies from ovipositing on their host fruits. Previous research found that kaolin was quite effective at reducing the number of *Rhagoletis pomonella* fruit flies (Walsh) visiting apples (*Villanueva & Walgenbach, 2007*). However, little is known about how efficient kaolin and other minerals reduce oriental fruit fly visitation, oviposition, and infestation in its host fruits.

The purpose of this study was to see how effective kaolin and other minerals are at controlling oriental fruit fly infestations in chili fruits.

Other minerals used in this study were considered to be relatively safe for humans and the environment. They also have insect control properties, as previously reported for bentonite (*El-Aziz, 2013*), calcium carbonate (*Morsi, 2021*), calcium hydroxide (*Estrada-Aguilar et al., 2012*), calcium oxide (*Smitha & Mathew, 2010*), dolomite (*Freitas et al., 2020*), sulfur (*Tacoli et al., 2020*), talc (*Driggers, 1929*), zeolite (*Floros et al., 2018*) and zinc oxide (*Gutiérrez-Ramírez et al., 2021*). We hypothesized that, in addition to kaolin, there would be other minerals effective against oriental fruit fly attacks.

# MATERIALS AND METHODS

## Location of study

This research was carried out in the Laboratory of Pesticides and Environmental Toxicology and the Experimental Field of the Faculty of Agriculture, Universitas Padjadjaran (UNPAD) Sumedang, West Java Province, Indonesia. The laboratory's temperatures ranged from 24 to 29 °C, with a 68% to 87% relative humidity. Temperatures in the experimental field ranged from 26 to 31 °C with a relative humidity of 70–82%.

## Oriental fruit fly

Adults of oriental fruit fly were housed in a 50 cm × 50 cm × 50 cm cage composed of iron and mesh. Sugar cubes, protein (yeast hydrolysate), and water were provided. An egging device comprised of a plastic cup (200 mL) filled with 100 mL of mango juice (Nutri sari) was used to harvest eggs. As a place for fruit flies to lay eggs, the top of the plastic cup was wrapped with plastic film (cling wrap). The eggs were washed under running tap water and placed on moistened tissue paper. Oriental fruit fly larvae were maintained in an artificial diet made from carrots (300 g), yeast (15 g), nipagin (1.5 g), propionic acid (4 mL), and water (250 mL). All ingredients were blended using a blender machine for 3 min and then placed on a disposable plastic plate. Under the plastic plate of larval diet, sand was placed for pupation. The pupae were collected (after filtering the sand), transported to a 1 L plastic container, and covered with moistened sand.

## Minerals

The effects of 10 minerals on oriental fruit fly infestation in chili fruits were studied. The minerals were obtained from PT Brataco Chemica (bentonite, calcium hydroxide, calcium oxide, kaolin, talc, zinc oxide), Kimia Mart (sulfur), PT Sakura Medica (calcium carbonate), Berkah Jaya Farm Shop (dolomite), and Kucingbilly Shop (zeolite). All minerals (in powders) were dried for 14 h at 60 °C in a drying oven. They were sieved twice through a 200-mesh sieve (74 microns) and again through a 400-mesh sieve (37 microns). Mineral formulations were created by combining minerals (80%; w/w) with a wetting agent (10%; w/w) and a dispersant agent (10%; w/w). Sodium lauryl sulfate was employed as a wetting agent, while sodium naphthalene sulfonate formaldehyde condensate was used as a dispersant agent.

## Laboratory experiment

Oriental fruit fly females (13–20 days old) were collected from the rearing cage 1 day before the test and placed in the experimental cage (20 females per cage) made of a plastic container (30 cm × 21 cm × 20 cm). Yeast hydrolysate and sugar cubes were fed to them. Water was sprayed on the inside interior of the experimental cage's walls. Ten mineral suspensions were made at the 2% (w/v) concentration. Twenty green chili fruits (10–12 cm) were immersed for 5 s in the mineral suspension. Twenty chili fruits were dipped in water as a control. After 30 min of air drying, the treated and control chili fruits were compared. Treated and control chili fruits were placed in separate experimental cages containing 20 oriental fruit fly females as part of a no-choice test. Chili fruits were placed on the bottom of the experimental cage with a gap of around 0.5–1 cm between them.

We counted the number of fruit flies on chili fruits every 15 min for 4 h. The average number was calculated by dividing the total number of fruit flies on chili fruits by the number of observations. Chili fruits were taken from the experimental cage after 24 h of exposure and counted for the number of oviposition punctures (holes), eggs laid, and infested fruits. The number of laid eggs was counted by dissecting the exposed chili fruits under a stereo microscope. If there was at least one fruit fly egg within a chili fruit, it was deemed infested by the oriental fruit fly. The experiment was run four times over multiple weeks. Each run of the experiment was considered as a blocking factor in a randomized block design (RBD).

## Semi-field experiment

Four minerals were chosen for further testing in a semi-filed test after significantly reducing oriental fruit fly visit and oviposition in the laboratory test. Twenty-four fruit fly cages were placed in an experimental field 1 day before the test. The cages were constructed of an iron frame (100 cm × 70 cm × 120 cm) with a white insect screen covering. Each experimental cage contains 20 oriental fruit fly females aged 13–20 days, given yeast hydrolysate, sugar cubes, and water. Mineral suspensions were made on the test day by mixing them with water at the concentration of 2% (w/v) and spraying them on a fruit-bearing chili plant grown in a plastic polybag. A synthetic insecticide emulsion (deltamethrin 25 g/L) was made at a concentration of 2 mL/L and sprayed on a fruit-bearing chili plant (positive control treatment). The untreated chili plant was used as a control. The treated chili plants were each exposed to the oriental fruit fly females within the experimental cage after air and sun drying for around 15 min. Chili fruits were collected and transported to the laboratory after 24 h of exposure. Chili fruits were initially examined for oviposition punctures on the skin before being dissected and examined under a stereo microscope for fruit fly eggs. If there was at least one fruit fly egg within a chili fruit, it was deemed infested by the oriental fruit fly. Each treatment was replicated four times and arranged in a Randomized Block Design.

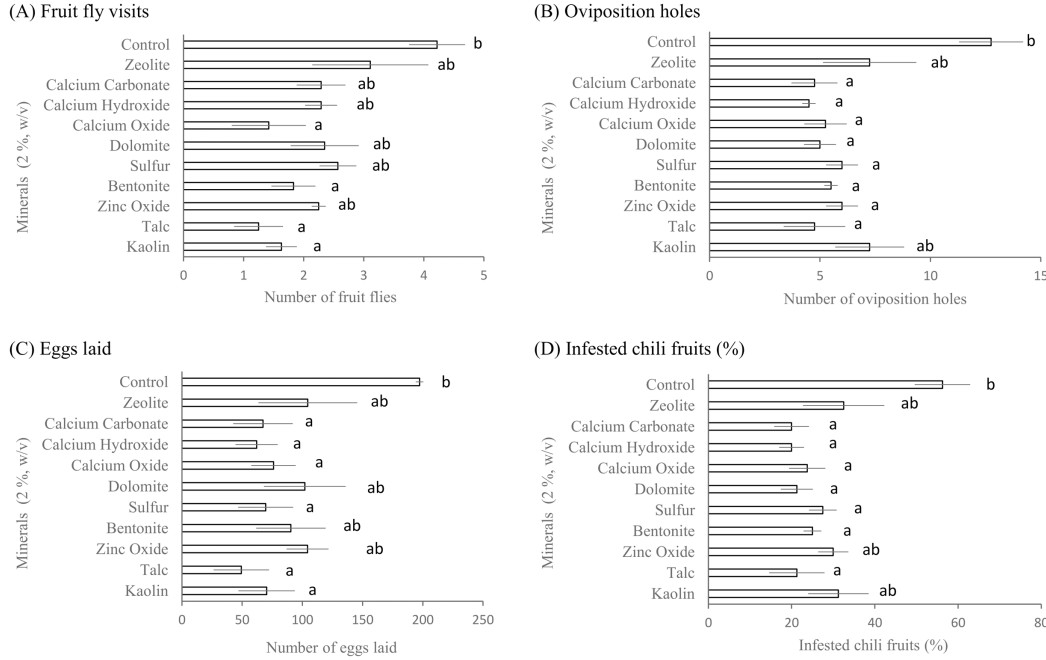

**Figure 1 Effects of some minerals on *B. dorsalis* under laboratory conditions.** Average (x̄ ± SE) numbers of (A) fruit fly visits, (B) oviposition holes, (C) eggs laid, and (D) % infested chili fruits. Fruit fly visits were recorded every 15 min for 4 h. Oviposition holes, eggs laid, and % infested chili fruits were recorded 24 h after treatment. Means followed by a different letter are significantly different at the 95% confidence level (Tukey's test: $P < 0.05$).

## Statistical analyses

Minitab version 16.0 was used to examine all of the data. Data were transformed when requirements for normality of data (Kolmogorov–Smirnov test) and homogeneity of variances (Levene test) were not satisfied. The data were analyzed with GLM ANOVA to see significant differences among treatments. The fixed factors were treatment (mineral) and run of the experiment (block). A further analysis was undertaken using the Tukey test at the 95% confidence level to identify treatments that had a significant difference.

## RESULTS

### Laboratory experiment

The effects of some minerals on the attacks of the oriental fruit fly on chili fruits were studied under laboratory conditions. Figure 1A shows the average number of oriental fruit flies visiting chili fruits. Applying kaolin, talc, bentonite, and calcium oxide to chili fruits resulted in significantly fewer oriental fruit flies than the control ($F_{10, 30} = 3.98$; $P = 0.002$). Other minerals had little effect on the number of oriental fruit flies visiting chili fruits. Oviposition holes (punctures) and deposited eggs were discovered in a laboratory test. The numbers of oviposition holes on chili fruits treated by talc, calcium hydroxide, calcium carbonate, calcium oxide, dolomite, bentonite, zinc oxide, and sulfur were significantly lower than the control ($F_{10, 30} = 4.19$; $P = 0.001$) (Fig. 1B). Only kaolin and zeolite had a negligible effect on the number of oviposition holes.

**Table 1 Average (x̄ + SE) number of fruit fly eggs and % infested chili fruits treated with some minerals 24 h after treatment under semi-field conditions.**

| Treatments | Laid eggs | Reduction in laid eggs (%) | % Infested chili fruits | Reduction in fruit fly infestation (%) |
|---|---|---|---|---|
| Talc (2%; w/v) | 115.75 + 8.45 c | 39.16 | 32.74 + 1.50 c | 40.97 |
| Calcium oxide (2%; w/v) | 69.25 + 15.54 b | 63.60 | 20.87 + 4.44 bc | 62.37 |
| Calcium hydroxide (2%; w/v) | 62.25 + 9.99 b | 67.28 | 18.40 + 3.54 b | 66.83 |
| Calcium carbonate (2%; w/v) | 98.00 + 10.96 bc | 49.49 | 22.88 + 4.18 bc | 58.74 |
| Deltamethrin 25 g/L (2 mL/L) | 0.00 + 0.00 a | 100.00 | 0.00 + 0.00 a | 100.00 |
| Control | 190.25 + 8.37 d | – | 55.46 + 7.16 d | – |

Note:
Means within a column followed by a different letter are significantly different at the 95% confidence level (Tukey's test: $P < 0.05$). Data of % infested chili fruits were arcsine transformed with ASIN(SQRT((X + 0.5)/100)) * 180/PI() prior to analysis of variance.

Compared to the control, the numbers of eggs laid by oriental fruit fly in chili fruits treated with kaolin, talc, sulfur, calcium oxide, calcium hydroxide, and calcium carbonate were significantly reduced ($F_{10, 30} = 2.90$; $P = 0.012$) (Fig. 1C). Zinc oxide, bentonite, dolomite, and zeolite were the only mineral formulations that had no significant effect on fruit fly oviposition.

The results also showed that the percentages of chili fruits infested by fruit flies were significantly lower in those fruits treated with talc, bentonite, sulfur, dolomite, calcium oxide, calcium oxide, and calcium carbonate than in those fruits of control ($F_{10, 30} = 3.64$; $P = 0.003$) (Fig. 1D). The reductions varied from 51.11% to 64.4%. On the other hand, other minerals (kaolin, zinc oxide, and zeolite) failed to significantly reduce the percentage of infested chili fruits compared to the control.

### Semi-field experiment

All of the minerals examined (talc, calcium oxide, calcium hydroxide, and calcium carbonate) significantly reduced the amount of *B. dorsalis* eggs laid in chili fruits in a semi-field bioassay ($F_{5, 15} = 57.91$; $P = 0.000$) (Table 1). Calcium oxide and calcium hydroxide were more effective than talc in controlling the number of eggs laid by *B. dorsalis* in chili fruits, but the two minerals were not significantly different from calcium carbonate. The application of deltamethrin 25 g/L at a concentration of 2 mL/L showed the highest efficacy of all treatments. Fruit fly oviposition in chili fruits was effectively inhibited by this insecticide, resulting in a 100% reduction in eggs laid. The four minerals tested, talc, calcium oxide, calcium hydroxide, and calcium carbonate significantly reduced the percentage of chili fruits infested by *B. dorsalis* ($F_{5, 15} = 56.50$; $P = 0.000$) in a semi-filed bioassay (Table 1). Calcium hydroxide was found to be substantially better than talc in controlling fruit fly infestation, but it did not differ significantly from calcium oxide or calcium carbonate. Despite this, the synthetic insecticide deltamethrin 25 g/L outperformed the four minerals, giving total protection against fruit fly infestation in chili fruits.

## DISCUSSION

Mineral effectiveness in controlling oriental fruit flies is poorly understood. In the current investigation, however, we found that some minerals could reduce the attack of this fruit fly both under laboratory and semi-field conditions. In a laboratory test, seven minerals (talc, bentonite, sulfur, dolomite, calcium oxide, calcium oxide, and calcium carbonate) significantly reduced the number of chili fruits infested by oriental fruit flies. The number of eggs laid in the treated chili fruits was dramatically reduced by five minerals (talc, sulfur, calcium oxide, calcium hydroxide, and calcium carbonate). Surprisingly, only three minerals (talc, bentonite, and calcium oxide) prevented oriental fruit fly visitation among the seven minerals significantly affected the number of infested chili fruits. Besides lowering the number of fruit flies visiting chili fruits, these findings show that minerals may have other bioactivities against oriental fruit flies.

Kaolin effectively reduced the number of visiting fruit flies and eggs laid, but it failed to reduce the number of infested chili fruits appreciably. Kaolin is the sole mineral that has been extensively researched for its ability against fruit fly attacks. Its formulations were effective for controlling fruit fly infestations (*Saour & Makee, 2004*; *Braham, Pasqualini & Ncira, 2007*; *Yee, 2008*; *Mezőfi et al., 2018*). Because of changes in its formulation/preparation as well as its concentration, kaolin has a reduced effect on oriental fruit fly infestation in this investigation. For example, we employed a concentration of 2%, whereas the study by *Mezőfi et al. (2018)* and *Braham, Pasqualini & Ncira (2007)* used concentrations of 4% and 5% (w/v), respectively.

In this investigation, the effects of zeolite and zinc oxide on the numbers of fruit flies visiting chili fruits, eggs laid, and infested chili fruits were not significant. On the other hand, these two minerals are effective against various insects. Zeolite was found to have a lethal effect on the adult bean weevils *Acanthoscelides obtectus* (*Floros et al., 2018*) and the sawtoothed grain beetle *Oryzaephilus surinamensis* (*Rumbos et al., 2016*). Therefore, zeolite could be used as grain protectants. Zinc oxide (nanoparticles) also had a strong mortality effect against whitefly *Trialeurodes vaporariorum* adults (*Khooshe-Bast et al., 2016*), rice weevil *Sitophilus oryzae* L. (*Haroun et al., 2020*), and cowpea beetle *Callosobruchus maculatus* F. (*Haroun et al., 2020*).

The minerals studied (talc, calcium oxide, calcium hydroxide, and calcium carbonate) reduced the number of eggs laid and infested chili fruits in a semi-field bioassay. Calcium hydroxide outperformed talc, although it was no better than calcium oxide or calcium carbonate. The present study is the first report on the effects of calcium oxide and calcium hydroxide against a fruit fly species. Nonetheless, calcium oxide has already been shown to be effective against the mealybug *Geococcus* spp. infestation on banana roots (*Smitha & Mathew, 2010*). Termites were also controlled by spraying a mixture of calcium oxide (1 kg) and copper sulfate (1 kg) in 100 L water on citrus trees (*Ashraf et al., 2014*). Similarly, calcium hydroxide has been shown to have a larvicidal effect on mosquitoes *Aedes aegypti* and *Culex quinquefasciatus* (*Estrada-Aguilar et al., 2012*), as well as the acaricidal effect on poultry red mites (*Hong et al., 2020*).

On the other hand, calcium carbonate is effective against fruit flies in a few studies. At a concentration of 5 kg/100 L, this mineral was found to reduce the attack of the peach fruit fly *Bactrocera zonata* Saunders on mangos (*Morsi, 2021*). According to *Ourique et al. (2019)*, a 20% concentration of liquid calcium carbonate reduced the total number of pupae and larvae of the South American fruit fly *Anastrepha fraterculus* (Wied.) in sweet oranges. The oviposition of the peach fruit moth, *Carposina sasakii* Matsumura, was similarly reduced by spraying calcium carbonate suspension on fruits (*Kazama et al., 2020*). Calcium carbonate was discovered to significantly reduce the population of mealy bugs infesting bananas (*Geococcus* spp.) when applied to the soil (*Smitha & Mathew, 2010*).

Physical factors that hinder a fruit fly from finding its host fruits could be one of the processes through which minerals reduce oriental fruit fly infestations. Fruit flies locate their host fruits by detecting the color, shape, and aroma of the fruits (*Hidayat et al., 2019*). However, after dipping chili fruits in a mineral suspension at a concentration of 2%, the color of the fruits did not alter significantly in this investigation. Therefore, it is possible if the reduced number of oriental fruit flies visiting the treated chili fruits is not related solely to the color of the treated chili fruits. The influence of fruit coating with mineral particles on fruit scent aroma is poorly understood. Nonetheless, other investigations found that kaolin particles applied to plant leaves did not impair leaf gas exchange (*Chamchaiyaporn et al., 2013*; *Nanos, 2015*).

Insect locomotion may be disrupted due to the physical effects of mineral particles. Talc powder, under dry condition, can provide a slick surface for adult black vine weevils, *Otiorhynchus sulcatus* (*F.*), preventing them from climbing, according to a study (*Bomford & Vernon, 2005*). A recent study also showed that calcium carbonate applied to apple fruits stopped the peach fruit moth *C. sasakii* from climbing up or settling on the fruits (*Kazama et al., 2020*). Similarly, *Salerno et al. (2020)* found that kaolin nanoparticle film significantly impacted the attachments of the Southern green stink bug *Nezara viridula* and the Mediterranean fruit fly *C. capitata*. Furthermore, mineral particles can cover the tarsal or footpads of adult insects, such as fruit flies, impairing their ability to move (*Boiteau & Vernon, 2001*).

Another theory, mineral particles had irritant effects on fruit flies. A previous study on using kaolin against a fruit fly found that it can operate as a tactile deterrent, lowering the time female *Rhagoletis pomonella* (Welsh) spends on treated fruits (*Leskey et al., 2010*). Another study discovered that the leafroller *Choristoneura rosaceana* (Harris) neonate larvae dispersed more quickly from plants treated with kaolin (*Sackett, Buddle & Vincent, 2005*).

Furthermore, several minerals have been reported to have a toxic effect on insects. Talc powder was shown to kill a significant percentage of juvenile oriental peach moth larvae (*Driggers, 1929*). The onion thrips, *Thrips tabaci* (Lind.) can likewise be killed by particle films made of kaolin and bentonite (*El-Aziz, 2013*). The apple maggot *Rhagoletis pomonella* (Walsh) (Diptera: Tephritidae) exposed to untreated surfaces lived significantly longer than those exposed to kaolin-treated surfaces in the presence of food and/or water, according to a study (*Leskey et al., 2010*).

In this experiment, each mineral was mixed with sodium lauryl sulfate (wetting agent) and sodium naphthalene sulfonate formaldehyde condensates (spreader). These two

surfactants are often used in agrochemical formulation (*Knowles, 1998*). The toxicity of sodium lauryl sulfate and sodium naphthalene sulfonate formaldehyde condensates to oriental fruit flies has never been studied before. We believe that the mineral component of the formulation was responsible for the reductions in the number of oriental fruit flies and oviposition on chili fruits observed in this study. The semi-field test revealed that calcium hydroxide was substantially more effective than talc at reducing fruit fly infestation. Because both calcium hydroxide and talc formulations contain the same amount of wetting agent and spreader, the variation in their anti-fruit fly effectiveness is most likely related to the mineral content of each formulation.

The use of calcium oxide, calcium hydroxide, or calcium carbonate at a concentration of 2% in semi-field bioassay must be combined with additional control procedures due to its medium level of control. It is intriguing to think about using these minerals as the push component of a push-pull strategy in controlling fruit fly infestation. Artificial fruit (*Hidayat et al., 2019*) and Ladd traps (*Schutze et al., 2016*) could be used as pull components. The principles of push-pull strategy in insect pest management and its potential components were previously described by *Cook, Khan & Pickett, 2007*.

Calcium oxide, calcium hydroxide, and calcium carbonate are considered safe for humans and the environment. These minerals are often used as a food additive (*Galvan-Ruiz, Banos & Rodriguez-Garcia, 2007*; *EFSA Panel on Food Additives & Nutrient Sources added to Food (ANS), 2011*). Nonetheless, the total dietary calcium consumption from all sources should not exceed 2,000 mg/day (*Galvan-Ruiz, Banos & Rodriguez-Garcia, 2007*). Farmers also regularly use soil amendments including calcium-based minerals to improve soil qualities for cultivation.

## CONCLUSION

Calcium oxide was one of ten minerals investigated as an alternate technique for the sustainable management of the oriental fruit fly on chili fruits. This mineral significantly reduced the numbers of visiting fruit flies, oviposition holes, and eggs laid, as well as the percentage of infested chili fruits in a laboratory test. However, in semi-field bioassay, calcium oxide and other minerals tested (calcium hydroxide, calcium carbonate, and talc) were less effective than the synthetic insecticide deltamethrin 25 g/L in controlling fruit fly infestation in chili fruits. Therefore, combining calcium oxide with other control approaches may result in superior control.

## ACKNOWLEDGEMENTS

Thanks to Agus Budi Hayono for his help in the semi-field experiment.

### Funding

This study was funded by the Ministry of Research, Technology and Higher Education of the Republic of Indonesia (Contract No: 007/UN6.E/LT/2019). The funders had no role in

study design, data collection and analysis, decision to publish, or preparation of the manuscript.

## Grant Disclosures
The following grant information was disclosed by the authors:
Ministry of Research, Technology and Higher Education of the Republic of Indonesia: 007/UN6.E/LT/2019.

## Competing Interests
The authors declare that they have no competing interests.

## Author Contributions
- Josua Crystovel Pangihutan performed the experiments, analyzed the data, prepared figures and/or tables, authored or reviewed drafts of the paper, and approved the final draft.
- Danar Dono conceived and designed the experiments, analyzed the data, authored or reviewed drafts of the paper, and approved the final draft.
- Yusup Hidayat conceived and designed the experiments, performed the experiments, analyzed the data, prepared figures and/or tables, authored or reviewed drafts of the paper, and approved the final draft.

## Data Availability
The raw data of laboratory and semi field studies are available in the Supplemental Files.

## Supplemental Information
Supplemental information for this article can be found online at http://dx.doi.org/10.7717/peerj.13198#supplemental-information.

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
