# Peer review of "The potency of minerals to reduce oriental fruit fly infestation in chili fruits"

_PeerJ, doi:10.7717/peerj.13198_

## Round 0.1 · original submission · Major Revisions

The manuscript has been reviewed by 3 expert referees who provide valuable comments.

Reviewer 1 ·

Basic reporting

No Comment

Experimental design

No Comment

Validity of the findings

No Comment

Additional comments

Line 25: damage

Line 29: Semi field I would refer to these at Semi field bioassays or cage bioassay throughout the manuscript.

Line 31: untreated control

Line 124-126: In the interest of reproducibility could sources of the minerals be included also?

Line 129-131: Did the control consist of a mixture of the wetting agent (Sodium Lauryl Sulfate) and the dispersant agent (Sodium Naphthalene Sulfonate formaldehyde condensates) or was it just water?

Line 171 - 172: What test was used to test normality and the homogeneity of variance?

Line 307 - 309: Great point!

Reviewer 2 ·

Basic reporting

The manuscript presents a study on response of fruit flies towards some minerals. The study is overall interesting and it will contribute to reinforce the valuable use of minerals in insect pest management. However, the work is poorly presented and not at all well described.
Introduction section is unnecessary long. Content presented in lines 53-62 is not required. The English language should be improved to ensure that an international audience can clearly understand your text (for example: line 102 is not clear). I recommend to ask an English native speaker to check the manuscript. It is better to mention the innovation of the study clearly.Figures and table have not labelled and described appropriately. Methods have not described with sufficient detail and information to replicate.

Experimental design

Original primary research within Aims and Scope of the journal.

Figures and table have not labelled and described appropriately. Methods have not described with sufficient detail and information to replicate.

Rigorous investigation performed to an ethical standard.

Validity of the findings

Impact and novelty not assessed. It is better to mention the innovation of the study clearly.

All underlying data have been provided; they are robust, statistically sound, & controlled.

Conclusions are well stated, linked to the original research question & limited to supporting results.

Additional comments

Some questions:
-Is chili correct? Based on some references like (Votava et al., 2005), I found chile for (Capsicum annuum var. annuum L.).
• Votava, Baral, and Bosland. 2005. Genetic diversity of Chile (Capsicum annuum var. annuum L.) landraces from Northern New mexico, Colorado, and Mexico. Economic Botany, 59, 8–17.
- Line 127-You dried all minerals at 60 °C for 14 hours. Why 14 hours? I could not find a reference in your manuscript.
- Line 138- In materials and methods section, you mentioned that twenty chilies were dipped in the mineral suspension with the concentration of 2% (W/V)? Why did you choose this concentration?
- Lines 139 and 161- You applied water as control and used minerals+wetting and dispersant agents for treatments. How to prove that efficacy in treatments is solely due to just minerals, and wetting and dispersant agents have no lethal effect?
-The titles of figures and table are not well described and should improve them. For example:
Average number of fruit fly visits on chili fruits (x̅ ± SE) ---> Average (x̅ ± SE) number of fruit fly visits on chili fruits treated with some minerals 24 hours after treatment under lab conditions.
Minor comments:
Line 29: in laboratory and in semi field ---> under laboratory and semi-field conditions
Line 31: in laboratory ---> under laboratory conditions
Line 46: Paticle film????
Line 50: Bactrocera dorsalis (Hendel) ---> Bactrocera dorsalis (Hendel) (Diptera: Tephritidae)
Line 65: one of important ---> one of the important
Line 83: Mediterranean ---> mediterranean
Line 84: Ceratitis capitata (Wiedemann) ---> Ceratitis capitata (Wiedemann) (Diptera: Tephritidae)
Line 88: spotted wing drosophila fruit fly Drosophila suzukii ---> spotted wing drosophila fruit fly, Drosophila suzukii
Line 91: boll weevil Anthonomus grandis---> boll weevil, Anthonomus grandis
Line 138: Twenty chili fruits??
Line 152: Semi field study---> Semi-field test
Lines 187, 192, 198, 200: Control---> control
Line 221: In the the ---> In the
Line 223: in laboratory and semi filed---> under laboratory and semi-field conditions
Line 243: efective ---> effective
Line 244: adult bean weevils ---> adult of bean weevils,
Line 256: calcium hydroxida ---> calcium hydroxide
Line 279: kondirion???
Line 295: It (Driggers, 1929) is a very old reference. Was there no newer reference?
Line 309: You suggested the application of calcium hydroxide, calcium oxide or calcium carbonate at the concentration of 2% need to be integrated with other control methods. I would like to know which methods?

Annotated reviews are not available for download in order to protect the identity of reviewers who chose to remain anonymous.

·

Basic reporting

Comments to the authors
The manuscript is interesting and have some merit but before accepting it for publication the following issues need to be addressed:

1 – There are some typo graphical errors that need to be corrected eg in line 73, 279 etc
2- The English language should be improved ,. Some examples where the language could be improved include lines 65, 71-73 , 134- ,178,the sentences should be properly rephrased for clarity
3. On the materials and methods- . The study location should be properly described on the first
paragraph .
- The name of the laboratory used for this study should be stated
- The laboratory conditions where the experiment was conducted should be clearly stated.
- Which type of sugar was used in feeding the adult fly? Specify.
-What percentage of sugar solution or quantity of sugar was used in feeding the
adult oriental fruit fly? Please specify
-The sources of minerals used must be clearly stated
. The author stated that he/she used Randomized Block Design in the laboratory study. - - What was the blocking effect? CRD is mostly used for laboratory studies, if otherwise, clarify by stating the blocking effects used

4. The title of figures 1-3 should be rephrased and the figure title should be placed under the
figure not above the figure
5. Authors should further provide proof that the used minerals are safe to man and environment
on the discussion section

Experimental design

The experimental design used for the laboratory studies need to be cross checked and justified if otherwise .

Validity of the findings

The findings are valid

Additional comments

No further comments

---

## Round 0.2 · Major Revisions

In addition to the comments provided by Reviewer 3, I ask the authors to consider the following:
• The manuscript contains a large number of grammatical errors. The authors should have their manuscript edited by a competent English speaker before resubmitting.
• Ten minerals were selected for testing in the laboratory experiment. What was the rationale for these treatment choices? Were there hypotheses associated with these treatments? (there should be) Hypotheses/predictions/rationale should be stated.
• Lab experiment not clear
o L145. It is not clear what data were collected every 15 min. Be specific. This appears to be a repeated measures scenario. How was aspect of the experimental design handled?
o Data were collected every 4 hour, but fruit were exposed to flies for 24 hour. Were the 15 min interval measurements done from 0-4 h, and then no further data were collected from 4-24 h?
o Exp design is not clear. Was time (e.g. each 15 min interval) the blocking factor within a run of the experiment, or was the run itself (i.e. repeating the experiment) considered the block?
• Field experiment
o L161. The purpose of the deltamethrin spray is not clear. Is this a positive control treatment? I assume so based on Table 1.
o L168. Not clear. Do you mean each treatment was replicated 4 times? Or that the experiment was repeated 4 times?
• Data analysis
o More detail needed on data analysis. State the fixed factors that were used in the GLM? Were there random factors in the experiment? This would necessitate use of a different model and analysis. E.g. there appears to have been a repeated measures element to the lab experiment.
o L172-73. What was done if conditions of normality and heterogeneity of error terms were not met?
• L177. Only two subheadings under Results are required: a subheading for “Laboratory experiment”, and a subheading for “Semi-field experiment”.
• L180. Fig 1 caption not correct. Were visits recorded for 24 h? On L144 you state visits were recorded every 15 min for 4 h. Should not the values in Fig 1 be representative of counts at 15 min intervals?
• L198. Fig 4 caption indicates percent infestation, not the number of infested fruit. Which is correct?
• Figs 1-4 can more effectively be combined into a single figure with 4 panels, rather than 4 separate figures.

Reviewer 2 ·

Basic reporting

Literature references, sufficient field background/context provided.
Professional article structure, figures, tables. Raw data shared.
Self-contained with relevant results to hypotheses.
Clear and unambiguous, professional English used throughout.

Experimental design

Original primary research within Aims and Scope of the journal.
Research question well defined, relevant & meaningful. It is stated how research fills an identified knowledge gap.
Methods described with sufficient detail & information to replicate.

Validity of the findings

All underlying data have been provided;
Conclusions are well stated, linked to original research question & limited to supporting results.

·

Basic reporting

Authors have made zealous efforts to revised the manuscripts based on the comments raised by the reviewers , however, I still observed minor errors that hat to be corrected before final publication of this manuscript
1. The two sentences in line 58 -59 should be combined as one sentence as ''These control techniques however, have limit as fruit fly trap still relies on the use of the male attractant (methyl eugenol)." not as written as two sentences.
2. In lines 122-126. authors tried to address the sources of the minerals as raised by putting it in the bracket. That indicated another meaning entirely. Since the ten minerals used were sourced from different places, It will be better to put in a tabular form to avoid clumsiness in describing the sources of each minerals in sentence form. The idea of putting it in bracket is out of place and it meant something else not what they intend to achieve.

Experimental design

OKAY

Validity of the findings

Valid

Additional comments

The manuscript should be considered for publication after the minor correction raised

---

## Round 0.3 · Minor Revisions

Here are a few minor corrections for you to consider:

1. L134-136 and elsewhere. No need to capitalize chemical names.

2. L150 and elsewhere. The term “visits” is a little ambiguous. I think what you did is counted the number of flies on fruit every 15 min for four hours. Phrasing like “… we counted the number of fruit flies on a chili fruit every 15 min…” is clearer.

3. L155-157. You are confusing the term “replicate” with “run”. Replicates refer to duplication of an experimental treatment. If you repeated the bioassay over multiple weeks, each of those repeats would be considered a ‘run’ of the experiment, which could be used, as you say, as a blocking factor.

4. L183. A run of the experiment would be a block

---

## Round 0.4 · accepted · Accept

You have adequately all my concerns and those of the reviewers.